# Danon Disease: Entire *LAMP2* Gene Deletion with Unusual Clinical Presentation—Case Report and Review of the Literature

**DOI:** 10.3390/genes14081539

**Published:** 2023-07-27

**Authors:** Adel Shalata, Marina Bar-Shai, Yarin Hadid, Muhammad Mahroum, Hila Mintz, Zaher Eldin Shalata, Evgeny Radzishevsky, Jacob Genizi, Avraham Lorber, Tamar Ben-Yosef, Liat Yaniv

**Affiliations:** 1The Simon Winter Institute for Human Genetics, Bnai Zion Medical Center, Haifa 32000, Israel; yarin.hadid@b-zion.org.il (Y.H.); mohammed.mahroum@b-zion.org.il (M.M.); hila.mintz@b-zion.org.il (H.M.); 2Ruth and Bruce Rappaport Faculty of Medicine, Technion-Israel Institute of Technology, Haifa 32000, Israel; evgeny.radzishevsky@b-zion.org.il (E.R.); jacob.genzi@b-zion.org.il (J.G.); avraham.lorber@gmail.com (A.L.); benyosef@tx.technion.ac.il (T.B.-Y.); liat.yaniv@b-zion.org.il (L.Y.); 3The Institute of Medical Genetics, Carmel Medical Center, Haifa 34362, Israel; marinaba1@clalit.org.il; 4Siba Rihana, Medical Center, Sakhnin 30810, Israel; zaher.shalata@gmail.com; 5Cardiology Department, Bnai Zion Medical Center, Haifa 32000, Israel; 6Department of Pediatric, Bnai Zion Medical Center, Haifa 32000, Israel; 7Pediatric Cardiology Unit, Rambam Medical Centre, Haifa 31096, Israel

**Keywords:** Danon disease, *LAMP2*-gene, genotype-phenotype correlation, penetrance, microdeletion, whole-exome-sequence

## Abstract

Danon disease is a rare x-linked dominant multisystemic disorder with a clinical triad of severe cardiomyopathy, skeletal myopathy, and intellectual disability. It is caused by defects in the lysosome-associated membrane protein-2 (*LAMP2*) gene. Numerous different mutations in the *LAMP2* protein have been described. Danon disease is typically lethal by the mid-twenties in male patients due to cardiomyopathy and heart failure. Female patients usually present with milder and variable symptoms. This report describes a 42-year-old father and his 3-year-old daughter presenting with mild manifestations of the disease. The father has normal intellectual development and normal physical activity. At the age of 13, he was diagnosed with mild ventricular pre-excitation known as Wolf–Parkinson–White syndrome (WPWs), very mild and mostly asymptomatic cardiomyopathy and left ventricular hypertrophy, and at about the age of 25 presented with visual impairment due to cone–rod dystrophy. His daughter showed normal development and very mild asymptomatic electrocardiographic WPWs abnormalities with left mild ventricular hypertrophy. Genetic testing revealed an Xq24 microdeletion encompassing the entire *LAMP2* gene. Relevant literature was reviewed as a reference for the etiology, diagnosis, treatment and case management.

## 1. Introduction

Danon disease (OMIM: 300257) is an x-linked dominant lysosomal storage disease characterized by the triad of cardiomyopathy, skeletal myopathy, and intellectual disability [1]. This disease is caused by defects in the lysosome-associated membrane protein 2 (*LAMP2*) gene, encoding for LAMP2 protein, which plays important role in autophagosome–lysosome fusion in autophagy [2,3].

The pathogenicity of *LAMP2* deficiency can be explained by the accumulation of autophagy materials [2,4], increased mitochondrial oxidative stress and increased apoptosis [5], altered metabolism, and impaired contractile function due to reduced calcium transients [2,4,5].

Typically, males are affected both earlier and more severely compared to females, with severe cardiomyopathy requiring heart transplantation; otherwise, the disease is lethal by their mid-twenties [6]. Additional clinical manifestations include retinal disease, hepatic disease, neurological manifestations, and pulmonary disease [6]. Currently, available treatments for Danon disease are limited to the prevention of sudden death and heart failure through a cardioverter defibrillator (implantable cardioverter-defibrillator, ICD) implantation and heart transplantation.

The most common clinical features in affected males include progressive hypertrophic cardiomyopathy (HCM), (in up to 96% of the patients) [6,7,8], accompanied by arrhythmias (in up to 80% of the patients), mild and slowly progressive proximal muscle weakness sometimes causing delayed motor developmental milestones (in up to 90% of the patients), progressive visual impairment due to retinopathy (in up to 20% of the patients) [9,10], and also mild intellectual disability (in up to 80% of the patients) [7]. Additional features may include elevated creatine kinase and hepatic enzymes, and a myopathic muscle biopsy [11]. Affected males usually develop heart failure, starting around 12 years old [12,13,14], and deteriorate to severe heart failure, almost certainly needing a transplantation, which occurs in the second and third decades of life [1,14]. Conduction abnormalities include complete heart block, atrial arrhythmias and life-threatening ventricular arrhythmias, pre-excitation, including Wolff–Parkinson–White syndrome (the most common finding on ECG), seen in approximately 48% of males at presentation [7,8,15]. Histologically, the myocard shows significant necrosis and fibrosis. HCM is a major determinant of clinical presentation and prognosis. The need for cardiac transplantation is inevitable for most males by the second to third decades of life [7,13,14,16]. Ventricular arrhythmias contribute significantly to sudden death in patients with Danon disease. Affected males, on average, live to age 19, while affected females live to an average age of 34 years [7,13,14,16].

Neuropsychiatric manifestations such as delayed speech, attention deficit disorder, behavioral problems, psychiatric disorders (e.g., severe depression, psychosis), autistic spectrum disorders, seizures, and axonal polyneuropathy with features of Charcot–Marie–Tooth disease have been described in many affected males (up to 69%) [7,17,18].

Some affected males may also exhibit hepatomegaly and elevated transaminases with preserved hepatic function, abdominal pain [7], diarrhea [7] and esophageal dysmotility in 77% of patients [7]. Respiratory disease is present in about half of affected males, with shortness of breath, chest tightness, coughing, and/or wheezing [7].

Females develop symptoms later than males, and usually have a cardiac-restricted phenotype presenting during adolescence or later by 19 years of age [12]. In affected females, either a hypertrophic (61–100% of female patients) or, less commonly, a dilated cardiomyopathy may develop. Up to 18% require cardiac transplantation, and arrhythmias occur in up to 100% of the patients. Atrioventricular block and sudden cardiac death due to arrhythmia have been reported in women. Relatively mild retinopathy has been reported in up to 20% of patients, mild non-progressive muscle weakness in up to 50% of patients, and mild intellectual disability in up to 10% of cases. Creatine kinase and hepatic enzymes may be normal or mildly elevated in affected females [12,13,14].

The treatment of Danon disease is mainly focused on the cardiac manifestations. A left ventricular assistance device may be implanted in order to provide sufficient cardiac output/bridge to transplantation and a cardioverted defibrillator in order to prevent sudden cardiac death from ventricular tachycardia [19]. Both male and female patients suffering from heart failure may benefit from heart transplantation. Five-year graft survival for male and female patients is relatively high, making heart transplantation an effective treatment option [19,20], although it appears that the long-term outcome in female patients is superior to that of males. Ablation of the arrhythmogenic focus may be considered in patients with cardiac pre-excitation and ventricular arrhythmia. There is no specific treatment for muscle weakness, but physiotherapy should be attempted and may become beneficial. Similarly, supportive interventions for developmental delay/intellectual disability, retinopathy, or neuropsychiatric manifestations should be offered. However, some preliminary evidence suggest that neuropsychiatric and intellectual function may improve after a successful heart transplantation [20,21]. A multidisciplinary team should follow these patients. Cardiac evaluation includes annual electrocardiography and echocardiography, a cardiac MRI every one to two years; ambulatory arrhythmia monitoring as needed based on symptoms; annual neurological assessment for muscle weakness and progression, monitoring of developmental progress, educational needs, and behavior at each visit with formal developmental assessments every three to five years during childhood; ophthalmologic evaluation at least every three to five years [1,13,22]. Genetic counseling to the patients and genetic testing of their relatives should be offered and undertaken [22].

As mentioned earlier, the underlying pathogenesis of Danon disease is the accumulation of lysosomal material secondary to *LAMP2* gene variants that disrupt LAMP2 protein proper function and autophagosome–lysosome fusion in autophagy [2,3,4,5].

*LAMP2* belongs to the LAMP protein family of highly homologous type I transmembrane proteins, each with a conserved glycosylated domain extending into the acidic lysosomal lumen. In humans, LAMP2 (CD107b) protein is derived from a single gene with nine coding exons and three alternate last exons, 9a, 9b, and 9c, generating three isoforms known as *LAMP2A, LAMP2B*, and *LAMP2C*, respectively, which differ primarily in the sequence of their transmembrane and cytosolic tail [16]. *LAMP2A* is a homodimer, with each monomer consisting of a large luminal domain, composed of two large domains (N-domain and C-domain separated by a hinge sequence), a transmembrane domain, and a C-terminal cytoplasmic domain [1] (Figure 1).

Each domain has its own distinct function:The N-terminal cytoplasmic domain of *LAMP2A* is composed of approximately 110 amino acids and contains several protein–protein interaction motifs, such as a WW domain, a PDZ-binding motif, and a dileucine motif. These motifs enable *LAMP2A* to interact with other proteins and participate in various cellular signaling pathways, including autophagy.The transmembrane domain of *LAMP2A* consists of a single α-helix that is approximately 25 amino acids long. It spans the lysosomal membrane and anchors *LAMP2A* in place. The transmembrane domain is amphipathic, with hydrophobic residues on one side that interact with the lipid bilayer of the membrane [23].The luminal domain of *LAMP2A* that faces the inside of the lysosome is composed of approximately 150 amino acids and contains several glycosylation sites. Glycosylation is the process by which sugar molecules are attached to proteins, and it plays a crucial role in proper folding, stability, and function of *LAMP2A*. The luminal domain also contains several conserved cysteine residues that form disulfide bonds, which help stabilize the protein [23].The C-terminal cytoplasmic domain of *LAMP2A* faces the cytoplasm and is responsible for interactions with cytoskeletal proteins and lysosomal trafficking. It is composed of approximately 60 amino acids and contains a proline-rich motif and a dileucine motif. The proline-rich motif enables *LAMP2A* to interact with cytoskeletal proteins, such as actin, and participate in lysosomal trafficking. The dileucine motif is recognized by adaptor proteins that mediate the transport of *LAMP2A* from the trans-Golgi network to the lysosome [23].

Overall, the different domains of *LAMP2A* enable the protein to interact with other proteins, anchor itself in the lysosomal membrane, participate in cellular signaling pathways, and mediate lysosomal trafficking. These functions are essential for proper lysosomal function and autophagy.

The three active transcripts of *LAMP2* gene encompass currently over 160 different published mutations [1,3] and more than 250 pathogenic and 28 likely pathogenic registered variants in well-acknowledged databases (LOVD (https://databases.lovd.nl/shared/genes/LAMP2, accessed on 15 May 2023), ClinVar (https://www.ncbi.nlm.nih.gov/clinvar, accessed on 15 May 2023) and Human Gene Mutation Database (HGMD) (https://www.hgmd.cf.ac.uk/ac/index.php, accessed on 15 May 2023) (Figure 1b, Appendix A).

The common variants in the *LAMP2* gene that disrupt LAMP2 protein function include (1) missense mutations, the substitution of a single amino acid in the *LAMP2* protein disrupts the formation of disulfide bonds and destabilizes the protein [7,24]; (2) nonsense, frameshift, splice, and deletion variants [7,24], which accordingly, create truncated or premature stop codons, non-functional proteins that are rapidly degraded [6,11,16,25].

Herein, two Danon patients are reported, a young male and his daughter, who both have an Xq24 microdeletion that includes the entire *LAMP2* gene and present with a mild disease phenotype.

## 2. Diagnostic Assessment—Methods and Results

### Patient Evaluation

This study was planned and executed by the Simon Winter Institute for human genetics’ team (Bnai-Zion Medical Center, Haifa, Israel). Informed consent was signed by Case 1 and by his wife for their affected child (Case 2). For initial evaluation, both parents and Case 1’s mother were interviewed. The medical records were reviewed, and a physical examination, chest radiography, electrocardiography (ECG), echocardiography, and molecular genetics tests were conducted. Left ventricular size and function were evaluated by M-mode and two-dimensional Doppler and color Doppler echocardiography, and cardiac arrhythmias were studied by 24-h Holter monitoring. In both the affected father and his wife and their daughter, a classical karyotype cytogenetics study was completed. In order to identify the specific responsible *LAMP2* gene variants, primers were designed to amplify all 9 *LAMP2* gene exons in Case 1. Then, a chromosomal microarray analysis (CMA) and whole-exome sequence (WES) were performed in both affected cases.

**CMA Procedure and Interpretation of Results**: Analysis of copy number variants (CNV) was performed as detailed elsewhere. An Affymetrix CytoScan750K CGH/SNP array was used to analyze DNA samples from the family according to the manufacturer’s protocol. DNA extracted from peripheral blood underwent digestion, ligation to adapters, amplification, and treatment with DNAse I. Fragments were end-labelled with a modified biotinylated base, hybridized to the array, and stained with a streptavidin-coupled dye and a biotinylated anti-streptavidin antibody. After scanning with a GeneChip Scanner, signal intensities were compared with reference reads using Chromosome Analysis Suite (ChAs V.4.0) software. Several public databases were routinely applied to characterize CNVs gains or losses, among them the UCSC Genome Browser (genome.ucsc.edu/), ClinVar (ncbi.nlm.nih.gov/clinvar/), the Database of Genomic Variants (dgv.tcag.ca/dgv/app/home), DECIPHER (decipher.sanger.ac.uk/), ISCA Consortium (iscaconsortium.org/), and ClinGen resource (clinicalgenome.org/). Following the American College of Medical Genetics and Genomics standards and guidelines for the interpretation and reporting of constitutional CNVs [26], CMA findings were classified as pathogenic, likely pathogenic, variants of unknown significance (VUSs), likely benign, and benign. Categorization was based on the original laboratory reports, which were reviewed according to the updated knowledge in the medical literature and the authors’ experience. Pathogenic and likely pathogenic findings (including both gross chromosomal aberrations and submicroscopic CNVs) were defined as clinically significant.

## 3. Whole-Exome Sequence

In Case 2, next-generation sequencing (NGS) was performed for whole-exome sequencing (WES) using a NovaSeq 6000 platform, with an S1 Reagent Kit v1.5 (200 cycles) (Illumina, San Diego, CA, USA) and an IDT xGen Exome Research Panel v2 (Integrated DNA Technologies, Coralville, IA, USA) for library preparation. The bioinformatics analyses focused on protein-altering variants (missense, nonsense, frameshift, and splice-site). Alignment and detection of variants were performed with DRAGEN bioinformatics pipeline version 3.6, with GRCh37 as the human genome reference. All called variants were assigned a quality value for filtering. Based on validation studies, the pipeline showed precision and detection > 99% for SNVs in areas with coverage greater than 20x and high mapping quality. Variants were annotated with the following public and internal bioinformatics/genetics resources: gnomAD, Ensembl Variant Effect Predictor (VEP), dbNSFP, Emedgene© DB, SnpSift, SnpEff, ExAC, GWAS, GRC, HGMD, Clinvar, Breast Cancer Information Core (BIC), BRCA Exchange, The Greater Middle East (GME), Amish Coriell and Mennonite, dbSNP, GERP, 1000 Genomes, Online Mendelian Inheritance in Man^®^ (OMIM), Clinical Genomic Database (CGD), Clinical Genomic Database (CGD), Orphanet, ClinVar, Emedgene© Gene-Disease Knowledge Graph.

Variants were also interpreted and classified using the ACMG/AMP/ClinVar standards and guidelines. Analysis for secondary findings was performed according to the current recommendations of the ACMG.

## 4. X Inactivation

An X inactivation pattern was determined at the Androgen Receptor (AR) gene locus using HhaI-sensitive methylation analysis: genomic DNA was PCR-amplified with primers flanking the polymorphic CAG repeat at the 5’ end of the AR gene after digestion with an HhaI methylation-sensitive restriction enzyme. PCR products were run on an ABI genetic analyzer and analyzed with GeneMapper software.

## 5. Results

### Case Description


**Case 1**


Initially, the healthy spouse of Case 1 was referred for genetic counselling in her sixth pregnancy with Case 2. She has healthy 7- and 5-year-old male children and a history of three spontaneous miscarriages, all of them occurring at the first trimester of spontaneous pregnancies. The family pedigree is presented below, depicting both affected and unaffected family members (Figure 2).

Her spouse complained of visual impairment secondary to retinitis pigmentosa as determined by ophthalmologic assessment, as described below.

For recurrent abortions study, peripheral blood chromosome analysis was performed in the couple, and a normal karyotype was found.

Case 1 was invited for further clinical and genetic evaluation, and a more complex clinical picture emerged. Today he is 42 years old, maintaining a full-time job as a storekeeper at a local industrial factory. Since the age of 25, he has reported progressive visual impairment with increased severity during the dark hours. Ophthalmic examination including visual evoked potentials (VEP) and an electroretinogram (ERG) studies revealed rods and cones dystrophy.

His healthy mother, who herself refused to undergo any genetic investigation, had reported that Case 1 complained of heart palpitations at the age of 13 years. According to his electrocardiogram (ECG) obtained then, an anomally appropriate with Wolf–Parkinson–White (WPW) pattern was described. However, his first echocardiography showed normal contraction of both ventricles and no signs of cardiomyopathy. Moreover, 24-h Holter ECG monitoring showed a short PR interval, multiple wide QRS complexes compatible with ventricular premature beats, and non-sustained ventricular tachycardia (VT) or supraventricular tachycardia (SVT) with an aberrant conduction, for which he was treated by amiodarone for several years. Basic metabolic evaluation (urine organic acids, plasma amino acids) was normal. Danon disease was considered by the pediatric team but further evaluation was not accomplished.

Of note, he has normal intelligence and no psychiatric, gastrointestinal, or pulmonary complaints. At the age of 18, despite the demonstration of increased echogenicity and borderline increase in antero-septal wall thickness which suggested a possible cardiomyopathy, he was drafted and was able to perform full combat warrant military service between 18 and 21 years of age without complications or physical limitations.

At the age of 27, a dual-chamber VDD (SENIA) cardiac pacemaker was implanted due to complete heart block. On recent evaluation, he has had atrial fibrillation and does not take medications. He reports good functional capacity, denies orthopnea, postural nocturnal dyspnea, palpitations, or syncope. However, he describes physical limitation only while running.

On physical examination, he has no signs of heart failure. His echocardiography (Figure 3) at age 39 showed normal ventricular size, moderate hypertrophy of the left ventricular wall, more prominent at the posterolateral segment than at the septum. Increased echogenicity of the myocardium with speckling was demonstrated, with a mildly thick aortic valve with minimal aortic regurgitation and mild mitral regurgitation. The left atrium is mildly dilated (28.3 cm^2^). Abnormal E/E’ was measured, supporting diastolic dysfunction. Representative echo mp4 video records can be found in Appendix A. He presented with a paced rhythm of 60 beats per minute with a wide QRS and atrial fibrillation (no A wave), and a pacing electrode seen in the right ventricle. However, in the original ECG recorded at 15 years old, a WPW pattern was demonstrated (Appendix A).

After his marriage in 2014, Case 1 participated in a research program aimed to investigate retinal disorders by NGS technology and WES analysis (PI Tamar Ben Yousef). The analysis was negative, and no pathogenic/likely pathogenic variants were found. The coverage was around 97%, and reading depth was X20. However, it is important to emphasize that the familiar WES platform and the bioinformatic tools used then (9 years ago) did not include CNVs detection in its routine analysis report. Taking into account this and his medical history (the combination of early cardiomyopathy, arrhythmia and conduction abnormalities, and ophthalmic involvement) that implied Danon disease (caused by *LAMP2* gene mutation) as the most appropriate diagnosis, we decided to focus on the *LAMP2* gene. We used the published primers [3] for direct Sanger sequencing. Compared to healthy control’s DNA sample (extracted from a peripheral blood sample), several *LAMP2* PCR analysis attempts of Case 1 DNA failed to retrieve any PCR products in all nine *LAMP2* exons. Reasonably, a suspicion of entire *LAMP2* gene’s deletion was raised. This assumption was confirmed by CMA analysis, indicating Danon disease (Figure 4). The mode of inheritance was discussed with his mother and his young sister. The sister underwent a CMA test, which was normal (two *LAMP2* alleles were seen), while his mother refused to be tested.


**Case 2**


The daughter of Case 1 was subsequently evaluated. Today, she is three years old, born after uneventful spontaneous pregnancy and delivery. During the pregnancy, the mother received two consequential genetic counselling sessions due to her history of recurrent miscarriages and her husband’s health condition. The first meeting was before and the second meeting was after the identification of *LAMP2* gene deletion in Case 1, supporting the diagnosis of Danon disease. The mother declined an amniocentesis procedure since; she intended to deliver the baby under any circumstances.

The family described a healthy and happy child with normal developmental progress. At the age of 14 months, she was examined by us in the genetic clinic as a routine follow-up and underwent karyotype and CMA analysis. As expected, she had inherited a normal maternal X-chromosome and a mutated paternal X-chromosome (harboring the *LAMP2* gene deletion). Therefore, she was a heterozygous carrier of a *LAMP2* deletion (Figure 4b). In the karyotype study, there was no evidence for chromosomal translocations or intra X-chromosome rearrangements. 

She was referred to a pediatric cardiologist at 18 months old and was diagnosed with heart murmur and asymptomatic WPW on baseline ECG (Figure 5).

She is thriving and is well developed with no heart-related symptoms. Her parents deny fatigue, syncope, or behavioral changes, and arrhythmia was not documented.

On recent clinical examination, she has no signs of congestive heart failure. A 1/6 systolic murmur was auscultated at the left sternal border. In the ECG, there was a normal sinus rhythm 139/min with high precordial voltage, a WPW pattern with a slightly widened QRS of 100 milliseconds with mild ST segment depression at the inferior and left lateral leads (Figure 5). Holter ECG showed a sinus rhythm of 85–167/min (average 119/min) with sparse atrial and ventricular ectopy. Some complexes were conducted with a narrow QRS and a different repolarization pattern and could represent competition between the accessory pathway and normal conduction.

The echocardiography showed normal ventricular size and normal contraction without wall motion abnormality. Increased echogenicity and hypertrophy involving the papillary muscles and posterolateral and apical segments was demonstrated. Valve morphology and estimated pulmonary pressure were normal, without pericardial effusion. The rest of the study was normal. Representative echo mp4 video records can be found in the Appendix A.

In summary: both cases have Danon disease due to *LAMP2* gene microdeletion (confirmed by PCR testing (in the father) and CMA studies (in both)). The father is hemizygous (Figure 4a) and his daughter is heterozygous for a 110.5Kb Xq24 microdeletion that includes the entire *LAMP2* gene (Figure 4b).

## 6. X-Inactivation Study

Although Case 2 presented with a mild phenotype, the early cardiac manifestation is uncommon in a female patient, and X-inactivation was tested, which may contribute to the understanding of Case 2’s cardiological findings. X-inactivation was tested by a restriction enzyme, sensitive for methylation of 5-prime of the Androgen Receptor (AR) gene [27]. The paternal/maternal alleles’ expression measured ratio of 35/65 (defined as normal) was compared with the 10/90 for X-inactivation ratio (the cutoff ratio for X-inactivation as defined by the by referral Lab of the Department of Medical Genetics, Hadassah Medical Organization).

Whole-Exome Sequence (WES): As described earlier, Case 2 was diagnosed with WPWS and mild hypertrophic cardiomyopathy. In order to investigate the early presentation independently, we decided to look for other genes involved in cardiomyopathy and arrythmia by WES analysis. An exome sequence is a technique used to analyze a person’s DNA sequence by sequencing only the protein-coding regions, comprising approximately 1% of the human genome, which contains ~180,000 exons split among ~22,000 different genes. The majority of known disease-causing mutations reside in these regions. To capture the exomic regions of DNA, a targeted capture method is used, followed by high-throughput sequencing using next-generation sequencing technologies. As described in the Methods section, a solo WES for Case 2 was run, and for analysis, the human phenotype ontology (HPO) “WPW syndrome” and “cardiomyopathy” was used following the ACMG standards and guidelines for the interpretation of sequence variants [26,28]. The sample quality pass was sufficient for clinical grade and precision and detection of >99% for SNVs in areas with coverage greater than 20× and high mapping quality as demonstrated in Figure 6.

The only pathogenic relevant variant found was a heterozygous deletion of 98.456 Kbp length in chromosome X (chrX:119504569—119603024(GRCh37/hg19) that included *ATP1B4* and *LAMP2* genes. In addition, one heterozygous missense variant of unknown significance (VUS) in the *TSC1* gene (chr9: 135786463G>T, *TSC1*, NM_000368.5: c.1067C>A, NP_000359.1: p.Thr356Asn) was identified. In silico tools predict this variant to be neutral. The conservation at this position is high, and splice prediction at this position is low. This variant has not been seen previously in our laboratory. The variant is present in ClinVar and absent from HGMD. This variant is reported in gnomAD (MAF 0). This variant is not present in the homozygous state in gnomAD. According to the ACMG guidelines, the variant is classified as uncertain significance. On physical examination, no signs of tuberous sclerosis were found, and a brain MRI is expected to provide more relevant findings. Another heterozygous VOUS incidental finding (chr8:145623065G>T, *CPSF1*, NM_013291.3:c.2103C>A, NP_037423.2:p.Tyr701Ter, Het) was found. The *CPSF1* gene is responsible for autosomal dominant Myopia-27 (MYP27), characterized by early-onset high myopia with increased axial lengths. An ophthalmic evaluation performed by a pediatric ophthalmologist determined mild myopia, and a routine follow-up was recommended

## 7. Discussion

Danon disease is an X-linked multisystem condition with predominant involvement of the heart, skeletal muscles, and retina, with overlying cognitive dysfunction [1]. It is caused by genetic mutations in the lysosome-associated membrane protein 2 (*LAMP2*) gene, resulting in *LAMP2* protein dysfunction [2,3]. Thus, the genetic defect disrupts cytoplasmic trafficking and leads to the accumulation of autophagic material and glycogen in skeletal and cardiac myocytes [2,3].

The vast majority of mutations in *LAMP2* are point mutations, of which NM_002294.3(*LAMP2*):c.928G>A (p.Val310Ile) is the most frequently encountered (described in five patients) (Figure 1a) [11].

This study reveals that an Xq24 microdeletion encompassing the entire LAMP2 gene is responsible for the Danon disease. However, multiple Xq24 microdeletions have been previously associated with Danon disease, hence reducing the novelty of this paper, as indicated by the following references [25,29,30]. Nevertheless, the unique clinical picture presented in this study opens excitedly the discussion about Danon phenotype expressivity. In 119 out of 126 pathogenic microdeletion cases reported in Clinvar in the *LAMP2* site (Figure 1b, Appendix A), the deletions include *LAMP2* and several other morbid genes. However, in the remaining seven reports, the deletions are relatively small and include the entire or part of the *LAMP2* gene [25,29]. The microdeletion mechanism appears to involve Alu-mediated unequal recombination and chromosomal breakage points involving TA-rich repeat sequences [30].

Although *LAMP2* variants with partial protein activity may be associated with a milder phenotype, there is no clear and consistent correlation between mutation type and phenotype. For example, the c.928G>A mutation, a highly variable phenotype described in the reported cases inclues learning and movement difficulties in infancy with a diagnosis of hypertrophic cardiomyopathy (HCM) at the age of 13; a typical triad of Danon disease; an unusual presentation of autism, and normal cardiac evaluation [11]. Further, Bottillo et al. [30] investigated the association between mutation types and clinical symptoms. They described that missense mutations were associated with a lower incidence of cardiomyopathy; truncation mutations had the earliest onset, followed by splicing mutations, and missense mutations had the latest onset, which is consistent with the findings of D’souza et al. [7]. Similarly, Fu L et al. [31] suggested that truncating mutations may lead to an earlier onset and more severe phenotypes [7,11,30,31].

The underlying mechanism for a pre-excitation abnormality may be linked to abnormal autophagy [15]. Danon-disease-patient-specific iPSC-based generated models have demonstrated that failure to clear the autophagic residue, rather than over-activation of the autophagy signaling pathway, is responsible for the accumulation of autophagy materials inside the affected cardiomyocytes [4,15]. Preliminary studies show that the correction of the autophagy defect ameliorates pathogenic pathways involved in Danon disease [4,32].

Currently, there are no available genetic nor biological therapies in Danon disease. Since the underlying mechanism of Danon disease is the blockade of autophagy, new drugs associated with autophagy, including autophagy activators and inhibitors, may be used in the near future to design new intervention strategies [15,32,33]. Moreover, in a recent study, Manso AM et al. [33] demonstrated the efficacy of a human *LAMP2B* gene transfer using a recombinant adeno-associated virus 9 carrying human *LAMP2B* (AAV9.*LAMP2B*) in a *LAMP2* knockout mouse, a Danon disease model. Hence, an intravenous AAV9 *LAMP2B* injection demonstrated a dose-dependent restoration of human *LAMP2B* protein in the heart, liver, and skeletal muscle tissue. Consequently, correction of the impaired autophagic flux (evidenced by increased LC3-II) was induced by a *LAMP2B* gene transfer in all tissues [33]. Future studies are needed to shed the light on the relevance of this strategy and its applicability in humans.

The mild phenotype in our affected 42-year-old male patient with entire *LAMP2* gene deletion is unexpected. Specifically, our male patient ended his fourth decade without any signs of heart failure and with mainly an ophthalmic phenotype of visual impairment mainly at night and dark spaces, explained by VEP and ERG studies, which revealed rods and cones dystrophy. To the best of our knowledge, this is the first such unique case. On the contrary, his daughter had her first cardiac evaluation at 18 months, and a pattern of WPWS in the ECG was found (Figure 5), accompanied with a normal cardiac echo study. Moreover, at 3 years of age, her cardiac echo study demonstrated increased echogenicity and hypertrophy involving the papillary muscles and posterolateral and apical segments ( mp4-Case-2). Of note, her WES analysis revealed a *LAMP2* deletion with no evidence for other pathogenic/likely pathogenic variants. If the *TSC1* gene VUS variant found will be classified as pathogenic in future, it may explain WPWS [34], but it cannot explain the papillary muscle and posterolateral and apical segments’ hypertrophy. Moreover, there was no evidence for cardiac rhabdomyoma generally found in tuberous sclerosis or any other typical disease features. Future studies are needed to clarify and elucidate this unpredicted clinical presentation in both.

The clinical heterogeneity in female patients could be attributed to the extent of an X-chromosome inactivation pattern (XCI), which occurs in female cells. When a random XCI occurs, the overlap of nuclear domains can rescue *LAMP2* expression in skeletal muscle fibers but not in cardiomyocytes (which do not regenerate), thus explaining why the majority of female patients developed cardiomyopathy but not skeletal myopathy. A random XCI pattern has been found in a variety of tissues from different patients [12,35] and different lysosomal storage diseases, including MPS-II and Fabry [33,36]. Conversely, when a skewed XCI favoring the mutant allele occurs, the majority of cardiomyocytes show *LAMP2* deficiency, and the overlap of nuclear domains in skeletal muscle is insufficient, resulting in early onset and severe cardiomyopathy and myopathy. A detrimental heterogeneous distribution of *LAMP2* (rather than an overall protein reduction) is crucial for the development of cardiomyopathy [35]. Eventually, our normal Case-2 XCI methylation analysis had excluded skewed XCI (meaning *LAMP2* alleles are supposed to be expressed equally). However, accurate preferential *LAMP2* gene expression can be evaluated in a muscle biopsy. It may be reasonable to think that despite a normal XCI study, the mutated allele was mainly expressed, and, accordingly, a diminished *LAMP2* gene expression predominates.

As mentioned earlier, the vast majority of Danon disease cases are due to point mutations in *LAMP2* gene. There are only a few reports of Xq24 microdeletions encompassing the *LAMP2* gene being the causative factor of Danon disease [29,36]. The penetrance in Danon disease is age-dependent and has not been well studied in the literature. Given the severity of the cardiac phenotype, the penetrance is estimated as high or nearly complete in males by the second decade [7].

In this study, the clinical picture of two related patients with an Xq24 microdeletion spanning the entire *LAMP2* gene was described. The male patient presented with a milder phenotypes of Danon disease, and his daughter with anearly onset phenotype. Together with the previous reports, they demonstrated that disease severity is more variable than was previously described. Thus, the Xq24 microdeletions can result in a milder clinical phenotype of Danon disease. These findings widen the understanding of the different mutation types leading to Danon disease and highlight the important value of the genetic testing in characterizing the phenotype and subsequent prognosis in patients with Danon disease.

**Study limitations:** Despite the comprehensive genetic evaluation, the exact underlying mechanism of this unique clinical picture of our Danon patients is still not fully understood. In the next paragraph, important points are raised in order to elaborate this unresolved issue.

In Danon disease, like in many other genetic disorders, disease penetrance and phenotype expressivity are variable [7,11,31,35,37]. This can be reflected by variable clinical presentation in individuals and familial cases having the same genetic variant [35,38,39,40,41,42,43]. Moreover, the genotype–phenotype correlation is well established in some genetic disorders, but no such correlation exists in others [38,39,40,41,42,43]. In Danon disease, there is controversy regarding the genotype–phenotype correlation [7,11,31,37].

It is reasonable to think that genetic/genomic variants (including coding and no-coding genes and intra- and intergenic alterations), epigenetics, environmental factors, and lifestyle [35] determine the gene expression profile which influences and determine protein–protein interactions and, consequently, contributes to the clinical picture and phenotypic heterogeneity and penetrance [35].

Moreover, in few genetic disorders the pseudogenes have an impact on phenotype severity. Spinal muscular atrophy (SMA) as a prototype is an excellent example, where, the majority of SMA patients are homozygous for *SMN1*-exons 7-8 deletion; however, phenotype severity is directly influenced by *SMN2* (*SMN1* pseudogene) copies that determine SMA subtype and can ameliorate the SMA phenotype [38]. Three paralogs, *LAMP1, LAMP2*, and *LAMP3,* are known. On the one hand, there is a pseudogene for *LAMP1* that was mapped to 12p13.3 [44], but, in contrast, there is no registered pseudogene for either *LAMP2* or *LAMP3.* Thus, the speculation that *LAMP2* deletion may be compensated by pseudogene expression remains theoretical. Moreover, the probability of compensation for *LAMP2* ablation by its paralogs is low. Hence, in a recent publication [44], the authors demonstrated that the expression of *LAMP1* and *LAMP2* is often associated with protection from lysosomal membrane permeabilization [44]. Meanwhile, *LAMP3* expression is associated with degradation of *LAMP1*, thus increasing lysosomal membrane permeabilization, inhibiting autophagy, and contributing to cell death; no such correlation between *LAMP2* and *LAMP3* was found [44].

Finally, the epigentics should be considered. In a Ng, K.M 2016 publication [4], it was demonstrated that downregulation of DNA methylation can ameliorate autophagy failure seen in Danon disease [4]. These findings encourage us to raise a question about the genomic DNA methylation (known as genomic episignature) profile of Case 1 patient compared with that of other Danon patients. We assume that his genomic episignature is be different and comparable to that found in mild patients rather than in severe Danon patients. This conjecture can be supported by plenty of publications in different genetic disorders [45,46,47].

The clinical phenotype was directly influenced by the episignature profile, and unique diagnostic episignature patterns were found in several entities, including neurodevelopmental disorders [45,46,47,48,49]. In conclusion, the phenotypic spectrum and genotype–phenotype correlations are strongly linked with the whole-genome methylation signature [48,49,50]. Future analyses with whole-genome sequencing, genomic epsignature, and gene expression profile in our Danon patients may contribute to further understanding.

## Figures and Tables

**Figure 1 genes-14-01539-f001:**
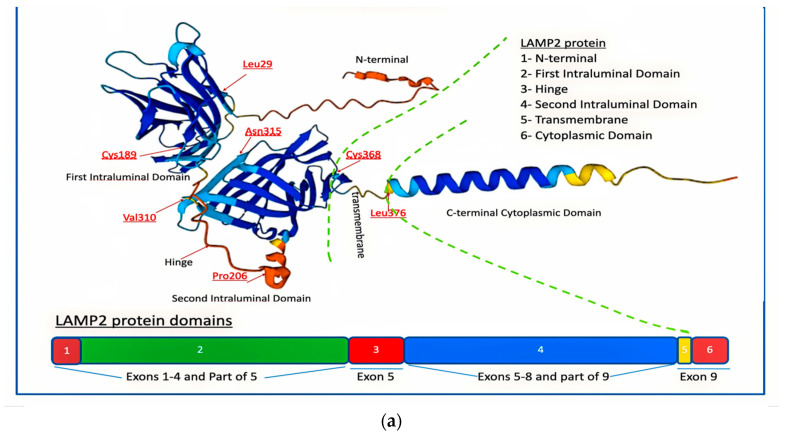
(**a**) LAMP2 protein and domains. Schematic representation of *LAMP2* protein’s predicted three-dimensional structure from AlphaFold, a courtesy of https://www.genecards.org accessible data. Some amino residues are depicted for illustration, and all others (except than 315Asn) are involved in Danon Disease. (**b**) Schematic illustration of HGMD *LAMP2* mutations according to variant subtypes (nonsense, missense, frameshift, splice, deletions) and their *LAMP2* exon location and distribution. The variant IDs in the specific location above exons are the key for the detailed molecular change as it appears in the Appendix A.

**Figure 2 genes-14-01539-f002:**
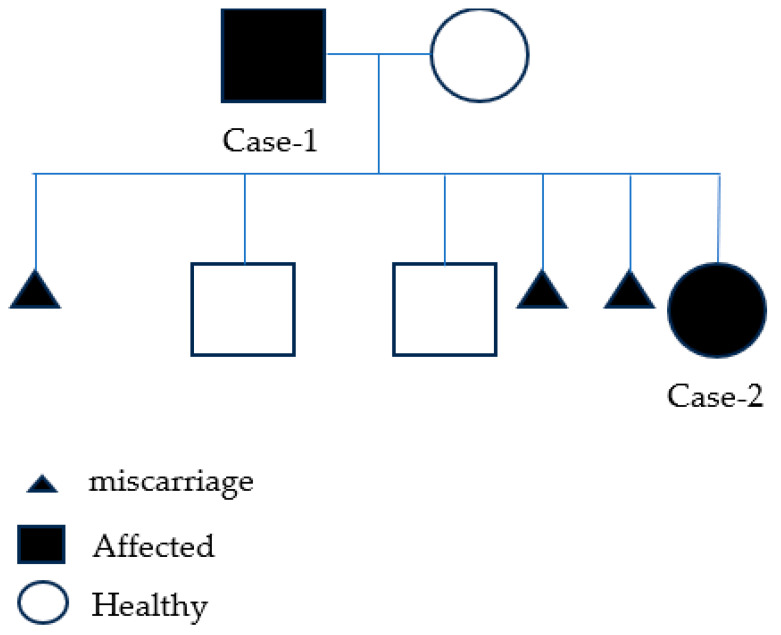
Family pedigree presenting the affected (filled) and the unaffected (no fill) family members, with three miscarriages.

**Figure 3 genes-14-01539-f003:**
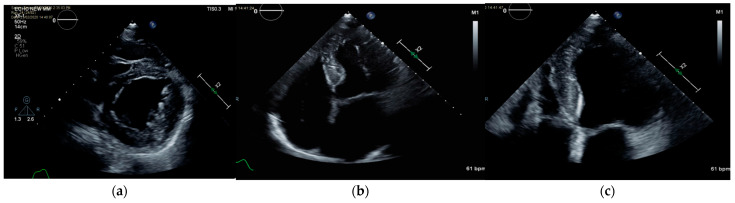
(**a**) Echocardiography of Case 1, short-axis view of the left ventricle showing mild hypertrophy, increased echogenicity, and normal contraction. (**b**) Four-chambers view (4cv) showing left ventricular (LV) hypertrophy with lateral wall involvement > septum, normal contraction, increased echogenicity, left atrial dilatation. (**c**) 4cv zoom of the left ventricle.

**Figure 4 genes-14-01539-f004:**
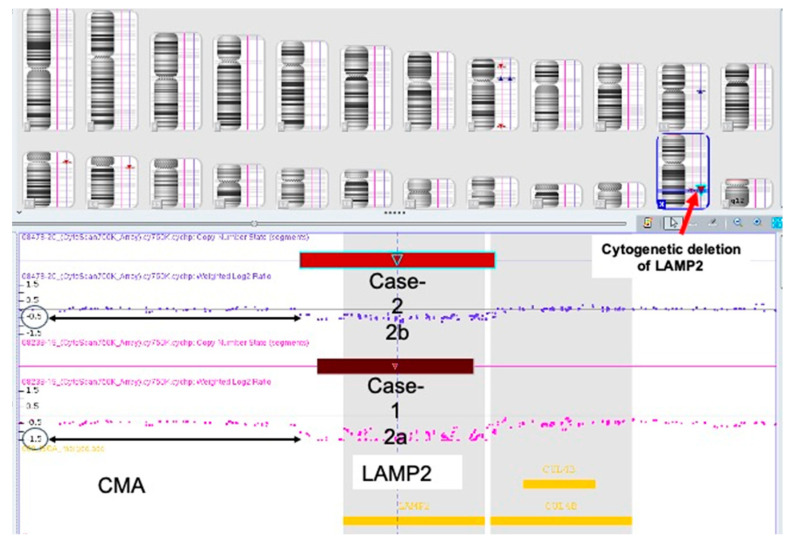
CMA analysis of Case 1 (**a**) and Case 2 (**b**), lower and upper blots, respectively. CMA of Case 1 identified deletion of 110.5 Kb on Xq24. (GRCh37)(Xq24(119513013_119623579)x0 (**a**). The same deletion was seen in a heterozygous state in his daughter (Case2) (**b**).

**Figure 5 genes-14-01539-f005:**
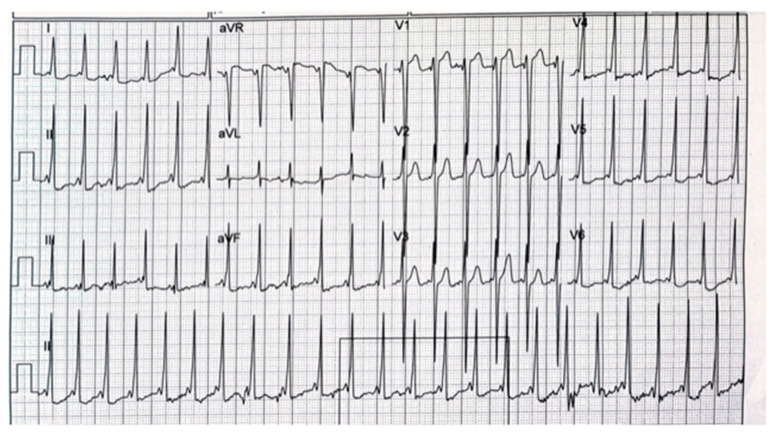
Case 2’s ECG results at the age of 18 months old, showing a WPW pattern.

**Figure 6 genes-14-01539-f006:**
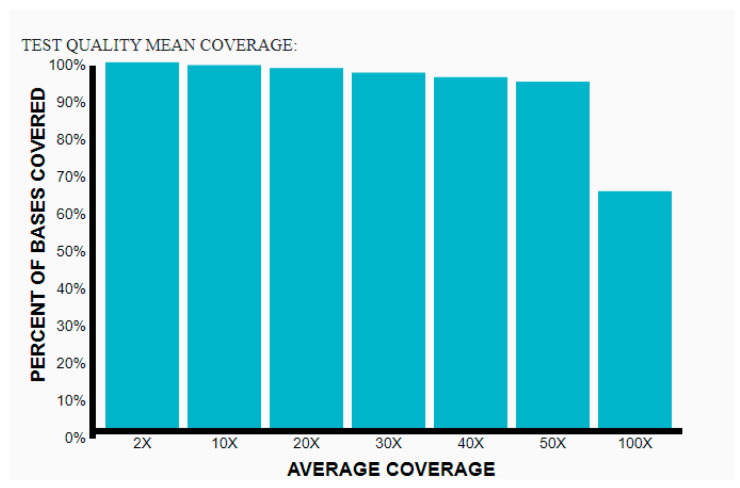
All called variants are assigned a quality value for filtering. Based on validation studies, the pipeline showed precision and detection > 99% for SNVs in areas with coverage greater than 20× and high mapping quality.

## Data Availability

All described data are provided upon request.

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
