# Peer review of "Danon Disease: Entire LAMP2 Gene Deletion with Unusual Clinical Presentation—Case Report and Review of the Literature"

_genes, 2023, doi:10.3390/genes14081539_

Round 1
Reviewer 1 Report
The authors present a case report involving a 42-year-old father and his 3-year-old daughter, both exhibiting mild manifestations of Danon disease, due to a Xq24 microdeletion that includes the entire LAMP-2 gene. This case offers valuable insights into the phenotypic expansion of Danon disease, shedding light on poorly understood aspects.
To enhance readability, it is recommended to present the case in a chronological order, with emphasis on the investigation of the daughter, followed by the father and possibly the grandmother. Including a heredogram would be beneficial for establishing the investigation process. Additionally, clarifying the reasons behind performing exome sequencing in the father and the potential importance of excluding an XXY pattern through karyotyping would be helpful.
The crucial question raised by this case is why the father only displayed mild manifestations of the disease. Unfortunately, the current Discussion section falls short of addressing this inquiry as it primarily focuses on the existing knowledge of Danon disease. To contribute significantly, the authors should concentrate the discussion on the challenges of explaining the milder effects observed in the affected father, even if it involves some speculative analysis.
Minor:
Can the authors please provide information about the classification and zygosity of the CPSF1 gene variant?
Furthermore, since the deletion involving LAMP-2 in case 1 was identified through CMA (chromosomal microarray), could you explain the rationale behind performing exome sequencing? Were there any expectations of identifying protective variants?
Additionally, it would be helpful if the authors could provide details regarding the methodological aspects of the exome sequencing, such as the mean coverage and the percentage of bases with coverage exceeding specific thresholds (e.g., >20x and >10x).
NC
Author Response
Thank you for reviewing our article.
We make our best to make the appropriate changes and additions to answer all the point and comments you had raised in your valuable review, and I will answer one by one:
1- To enhance readability, it is recommended to present the case in a chronological order, with emphasis on the investigation of the daughter, followed by the father and possibly the grandmother. Including a heredogram would be beneficial for establishing the investigation process. Additionally, clarifying the reasons behind performing exome sequencing in the father and the potential importance of excluding an XXY pattern through karyotyping would be helpful.
We make our best to organize the medical history their data to be presented in chronological manner.
As described in results section, the mother of Case-1 refused to participate in genetic testing.
Case-1 karyotype was 46XY, normal. The Karyotype in both couple was performed as routine workup to investigate recurrent miscarriages.
Since Case-2 (daughter of Case-1) presented WPWS and cardiomyopathy at age 1-2 years old, we completed NGS sequencing in order to see if she had other mutation in other genes involved in cardiomyopathy.
Case-1 performed WES 9 years earlier in attempt to explain his RP and find the responsible mutated gene. See more details please in Described in methods and results.
2.
The crucial question raised by this case is why the father only displayed mild manifestations of the disease. Unfortunately, the current Discussion section falls short of addressing this inquiry as it primarily focuses on the existing knowledge of Danon disease. To contribute significantly, the authors should concentrate the discussion on the challenges of explaining the milder effects observed in the affected father, even if it involves some speculative analysis.
We do agree and thank you for your important comment. We added our thoughts in the discussion section and tried to explain the unusual presentation although, theoretically.
Minor:
Can the authors please provide information about the classification and zygosity of the CPSF1 gene variant?
Answer:
Heterogeneous (it's mentioned in the result WES section)
done
Furthermore, since the deletion involving LAMP-2 in case 1 was identified through CMA (chromosomal microarray), could you explain the rationale behind performing exome sequencing? Were there any expectations of identifying protective variants?
Answer:
Since Case-2 (daughter of Case-1) presented WPWS and cardiomyopathy at age 1-2 years old, we completed NGS sequencing in order to see if she had other mutation in other genes involved in cardiomyopathy.
Additionally, it would be helpful if the authors could provide details regarding the methodological aspects of the exome sequencing, such as the mean coverage and the percentage of bases with coverage exceeding specific thresholds (e.g., >20x and >10x).
We add the requested information in result section and added figure 7 to represent the coverage and depth of reading.
Study limitations: Despite the comprehensive genetic evaluation, the exact underlying mechanism of this unique clinical picture of our Danon patients is still not fully understood. In the next paragraph we will raise important points in order to elaborate this unresolved issue.
In Danon disease, like in many other genetic disorders, the disease penetrance and phenotype expressivity are variable [7,11,31,35,37]. This can be reflected by variable clinical presentation in individuals and familial cases having the same genetic variant [35,38–43]. Moreover, the genotype-phenotype correlation is well established in some genetic disorders while, no such correlation exists in others [38–43]. In Danon disease controversy exist regarding the genotype-phenotype correlation [7,11,31, 37].
It is reasonable to think that genetic/genomic variants (including coding and no-coding genes and intra- and intergenic alterations), epigenetics, environmental factors and lifestyle [35] determine the gene expression profile which influence and determine the protein-protein interaction and consequently, contribute to clinical picture and phenotypic heterogeneity and penetrance [35].
Moreover, in few genetic disorders the pseudogenes have an impact on phenotype severity. Spinal muscular atrophy (SMA) as a prototype is an excellent example, where, the majority of SMA patients are homozygous for SMN1-exons 7-8 deletion however, the phenotype severity is directly influenced by SMN2 (SMN1 pseudogene) copies which determine SMA subtype and can ameliorate SMA phenotype [38]. Three paralogs, LAMP1, LAMP2 and LAMP3, are known. On one side, there is a pseudogene for LAMP1 that was mapped to 12p13.3 [44], while, in contrast, there is no registered pseudogene for either LAMP2 or LAMP3. Thus, the speculation that LAMP2 deletion may be compensated by pseudogene expression remains theoretical. Moreover, the probability of compensation for LAMP2 ablation by its paralogs is low. Hence, in a recent publication [44], the authors demonstrated that the expression of LAMP1 and LAMP2 is often associated with protection from lysosomal membrane permeabilization [44]. [While, LAMP3 expression is associated with degradation of LAMP1 thus, increasing lysosomal membrane permeabilization, inhibiting autophagy and contributing to cell death; no such correlation between LAMP2 and LAMP3 was found [44].
Finally, the epigentics should be considered. Thus, in Ng, K.M 2016 publication [4], it was demonstrated that downregulation of DNA methylation can ameliorate autophagy failure seen in Danon Disease [4]. These findings encourage us to raise a question about genomic DNA methylation (known as genomic episignature) profile of Case-1 patient compared with other Danon patients. We assume that his genomic episignature supposes to be different and comparable to that found in mild patients more than to severe Danon patients. This conjecture can be supported by plenty of publications in different genetic disorders [45–47].
Where, the clinical phenotype was directly influenced by the episignature profile, and unique diagnostic episignature patterns were found in several entities including neurodevelopmental disorders [45–47] [48,49]. In conclusion, the phenotypic spectrum and genotype-phenotype correlations are strongly linked with whole genome methylation signature [48–50]. And future analyzes by whole genome sequence, genomic epsignature and gene expression profile in our Danon patients may shed lights and contribute to our understanding.

Reviewer 2 Report
The article describes a clinical case of a family with Danon's disease and provides a literature review on the genetic and phenotypic correlations of this disease. Research methods are modern and adequate. Literature review is interesting and quite informative. At the same time, the article does not give a clear answer about the cause of the non-severe phenotype in a proband with the absence of the LAMP2 gene. And this should be reflected in the "limitations of the study" section. The name of the gene in the article should be written in italics (for example - LAMP2)
Author Response
We thank you for your valuable comments
We make, hopefully, the requested corrections.
1. We make English minor corrections
2. Study limitations:
We do agree and thank you for your important comment. We added our thoughts in the discussion section and tried to explain the unusual presentation although, theoretically.
We added the following to the discussion:
Study limitations: Despite the comprehensive genetic evaluation, the exact underlying mechanism of this unique clinical picture of our Danon patients is still not fully understood. In the next paragraph we will raise important points in order to elaborate this unresolved issue.
In Danon disease, like in many other genetic disorders, the disease penetrance and phenotype expressivity are variable [7,11,31,35,37]. This can be reflected by variable clinical presentation in individuals and familial cases having the same genetic variant [35,38–43]. Moreover, the genotype-phenotype correlation is well established in some genetic disorders while, no such correlation exists in others [38–43]. In Danon disease controversy exist regarding the genotype-phenotype correlation [7,11,31, 37].
It is reasonable to think that genetic/genomic variants (including coding and no-coding genes and intra- and intergenic alterations), epigenetics, environmental factors and lifestyle [35] determine the gene expression profile which influence and determine the protein-protein interaction and consequently, contribute to clinical picture and phenotypic heterogeneity and penetrance [35].
Moreover, in few genetic disorders the pseudogenes have an impact on phenotype severity. Spinal muscular atrophy (SMA) as a prototype is an excellent example, where, the majority of SMA patients are homozygous for SMN1-exons 7-8 deletion however, the phenotype severity is directly influenced by SMN2 (SMN1 pseudogene) copies which determine SMA subtype and can ameliorate SMA phenotype [38]. Three paralogs, LAMP1, LAMP2 and LAMP3, are known. On one side, there is a pseudogene for LAMP1 that was mapped to 12p13.3 [44], while, in contrast, there is no registered pseudogene for either LAMP2 or LAMP3. Thus, the speculation that LAMP2 deletion may be compensated by pseudogene expression remains theoretical. Moreover, the probability of compensation for LAMP2 ablation by its paralogs is low. Hence, in a recent publication [44], the authors demonstrated that the expression of LAMP1 and LAMP2 is often associated with protection from lysosomal membrane permeabilization [44]. [While, LAMP3 expression is associated with degradation of LAMP1 thus, increasing lysosomal membrane permeabilization, inhibiting autophagy and contributing to cell death; no such correlation between LAMP2 and LAMP3 was found [44].
Finally, the epigentics should be considered. Thus, in Ng, K.M 2016 publication [4], it was demonstrated that downregulation of DNA methylation can ameliorate autophagy failure seen in Danon Disease [4]. These findings encourage us to raise a question about genomic DNA methylation (known as genomic episignature) profile of Case-1 patient compared with other Danon patients. We assume that his genomic episignature supposes to be different and comparable to that found in mild patients more than to severe Danon patients. This conjecture can be supported by plenty of publications in different genetic disorders [45–47].
Where, the clinical phenotype was directly influenced by the episignature profile, and unique diagnostic episignature patterns were found in several entities including neurodevelopmental disorders [45–47] [48,49]. In conclusion, the phenotypic spectrum and genotype-phenotype correlations are strongly linked with whole genome methylation signature [48–50]. And future analyzes by whole genome sequence, genomic epsignature and gene expression profile in our Danon patients may shed lights and contribute to our understanding.
Reviewer 3 Report
This study reveals that an Xq24 microdeletion encompassing the entire LAMP-2 gene is responsible for the Danon disease. However, multiple Xq24 microdeletions have been previously associated with the Danon disease, hence reducing the novelty of this paper, as indicated by the following references. [1] Yang Z, Funke BH, Cripe LH, Vick GW 3rd, Mancini-Dinardo D, Peña LS, Kanter RJ, Wong B, Westerfield BH, Varela JJ, Fan Y, Towbin JA, Vatta M. LAMP2 microdeletions in patients with Danon disease. Circ Cardiovasc Genet. 2010;3(2):129-37. doi: 10.1161/CIRCGENETICS.109.901785. [2] Majer F, Piherova L, Reboun M, Stara V, Pelak O, Norambuena P, Stranecky V, Krebsova A, Vlaskova H, Dvorakova L, Kmoch S, Kalina T, Kubanek M, Sikora J. LAMP2 exon-copy number variations in Danon disease heterozygote female probands: Infrequent or underdetected? Am J Med Genet A. 2018;176(11):2430-2434. doi: 10.1002/ajmg.a.40430. [3] Majer F, Kousal B, Dusek P, Piherova L, Reboun M, Mihalova R, Gurka J, Krebsova A, Vlaskova H, Dvorakova L, Krihova J, Liskova P, Kmoch S, Kalina T, Kubanek M, Sikora J. Alu-mediated Xq24 deletion encompassing CUL4B, LAMP2, ATP1B4, TMEM255A, and ZBTB33 genes causes Danon disease in a female patient. Am J Med Genet A. 2020;182(1):219-223. doi: 10.1002/ajmg.a.61416.Minor editing of English language is required.
Author Response
1- We rewrite the introduction
We moved and organized paragraphs from the discussion part to the introduction.
2-English language editing- performed
3- We took your suggestion and added to the discussion with our addition (highlighted)
[This study reveals that an Xq24 microdeletion encompassing the entire LAMP-2 gene is responsible for the Danon disease. However, multiple Xq24 microdeletions have been previously associated with the Danon disease, hence reducing the novelty of this paper, as indicated by the following references [25,29, 30]. ] Nevertheless, the unique clinical picture presented in this study opens excitedly the discussion about Danon phenotype expressivity.
Round 2
Reviewer 3 Report
My suggestion is Accept in present form, but the references should be formatted (in reference 2, the journal name is missing ).